# Vitamin D Levels in Children with Recurrent Acute Tonsillitis in Jordan: A Case-Control Study

**DOI:** 10.3390/ijerph19148744

**Published:** 2022-07-18

**Authors:** Baeth Moh’d Al-Rawashdeh, Mohammad Altawil, Fareed Khdair Ahmad, Abdelrahman Alharazneh, Lubna Hamdan, Ahmed S. H. Muamar, Sireen Alkhaldi, Zahraa Tamimi, Rawand Husami, Raihan Husami, Nidaa A. Ababneh

**Affiliations:** 1Department of Otolaryngology, School of Medicine, University of Jordan, Amman 11942, Jordan; amuammar1991@gmail.com; 2Department of Otolaryngology, South Infirmary Victoria University Hospital, T12 X23H Cork, Ireland; mohammad.al-tawil@hotmail.com; 3Department of Pediatrics, School of Medicine, University of Jordan, Amman 11942, Jordan; f.khdair@ju.edu.jo; 4Department of Otolaryngology, School of Medicine, Mutah University, Al-Karak 61710, Jordan; haraznehm@mutah.edu.jo; 5Department of Pediatric Infectious Diseases, Vanderbilt University Medical Center, Nashville, TN 37232, USA; lubna.r.hamdan@vumc.org; 6Department of Public Health, University of Jordan, Amman 11942, Jordan; s.alkhaldi@ju.edu.jo; 7School of Medicine, University of Jordan, Amman 11942, Jordan; zahtam@gmail.com (Z.T.); rawandhusami13@gmail.com (R.H.); raihan555@hotmail.com (R.H.); 8Cell Therapy Center, University of Jordan, Amman 11942, Jordan; n.ababneh@ju.edu.jo

**Keywords:** vitamin D, vitamin D deficiency, recurrent tonsillitis, children

## Abstract

Background: Vitamin D is essential for many functions of the body. In addition to its primary function of regulating the absorption of calcium in the small intestine, its role in the immune system has recently been studied. The current study aimed to test the impact of vitamin D deficiency on the rate of recurrent acute tonsillitis in children. Methods: According to Paradise criteria, two hundred forty-two children with recurrent acute tonsillitis were recruited. A group of healthy children (*n* = 262) was also recruited as controls. Poisson regression was run to predict the number of tonsillitis episodes per year based on vitamin D levels. The mean vitamin D level in the study group was lower than in the control group (*p* < 0.0001). Poisson regression of the rate of recurrent tonsillitis and vitamin D level (OR = 0.969 (95% CI, 0.962–0.975)) showed that for every single unit increase in vitamin D level, there was a 3.1% decrease in the number of tonsillitis episodes per year (*p* < 0.0001). Conclusions: Vitamin D deficiency is associated with higher rates of recurrent acute tonsillitis. Future controlled trials should investigate the role of vitamin D supplementation in reducing the rate of recurrent tonsillitis.

## 1. Introduction

Vitamin D is a fat-soluble vitamin, which is involved in many essential functions of the human body, including calcium and phosphate homeostasis. Amongst other functions, vitamin D has been recently studied for its role in the immune system. Vitamin D has been found to modulate the innate and adaptive immunity, as well as the inflammatory cascade. Vitamin D receptor (VDR) is expressed by many immune cells, suggesting its role in immune modulation and inflammation [1,2,3,4]. The principal circulating form of vitamin D is 25-hydroxyvitamin D (25(OH)D), and this is considered the best indicator for vitamin D levels from both endogenous and dietary sources [5].

Vitamin D deficiency is linked [1] to an increased rate of respiratory tract infections (RTIs) in children [5,6] and adults [7,8]. Children having nutritional rickets have demonstrated rachitic lung due to respiratory tract infections [9]. Vitamin D deficiency has also been associated with other infections, cancers, and cardiovascular diseases [10]. Various randomized controlled trials have concluded that vitamin D supplementation reduce the rate of upper respiratory tract infections (URTIs) [11], while others were inconclusive [12].

The palatine tonsils play a prominent role in recognizing airborne antigens, with the help of macrophages found in their crypts [13]. Toll-like receptors (TLR), which are located on the macrophages, bind to the antigens taken up by the epithelial cells of the tonsillar crypts, thus helping in differentiating the structural molecules of the pathogens from the eukaryotes [14]. The expression of the TLR becomes variable in the case of recurrent tonsillitis [15]. The activation of sTLR increases the expression of the VDR gene and vitamin D hydroxylase required for vitamin D synthesis [16]. The macrophages use extracellular 25(OH)D to synthesize 1,25-dihydroxyvitamin D (1,25(OH)2 D) [17], which binds to the VDR and stimulates the synthesis of antimicrobial peptides (AMPs), such as defensin and cathelicidin [17,18]. AMPs are the main defense factors in the upper respiratory tract (URT) and play a important role in toxin neutralization and chemotactic activity [3,4,16,18,19]. Treating macrophages with 1,25(OH)2 D in vitro increased the production of cytokines, including IL-6, IL-12, and TNF-α [20,21].

This study compared vitamin D levels in children having recurrent tonsillitis to a control group. It also evaluated vitamin D deficiency (<20 ng/mL) as a risk factor for recurrent tonsillitis. We also investigated the possible associations between vitamin D levels and the demographic and clinical characteristics of children with recurrent tonsillitis.

## 2. Materials and Methods

### 2.1. Study Design

This observational case-control study was conducted from January 2017 through January 2020 at the Hospital of the University of Jordan, a tertiary referral teaching hospital in Jordan, abiding by the Helsinki Declaration (Ethical Principles for Medical Research Involving Human Subjects). The study was approved by the IRB and ethical committee of the University of Jordan (IRB No: (120/2014/IRB/L). Each participant’s parent signed an informed consent describing the nature of the study, contact information, protection of privacy, and confidentiality. It contained the essential details to be addressed by the participants’ parents, to decide on their children’s participation, including potential risks and the benefits of the study for the participants and the community. Their right to withdraw from the study at any time was explained. Participants’ confidentiality and privacy were ensured using a secured computerized system for data entry and analysis, which was accessible only to the study team. Blood samples for the study and control groups were obtained in the ENT and pediatric clinics, after ensuring that the participants had met the eligibility criteria and signed the informed consent.

### 2.2. Subjects

Children aged between 2 to 14 years with recurrent tonsillitis were recruited for the study (Group 1). Group 1 included 253 children who met the inclusion criteria, but 11 blood samples were inadequate for vitamin D testing, so they were excluded, and only 242 patients were included in the analysis.

#### 2.2.1. Inclusion Criteria for Group 1

Fulfilling the Paradise criteria for recurrent tonsillitis, of at least seven episodes of tonsillitis in the previous year, at least five episodes per year in the previous two years, or at least three episodes per year in the previous three years [22].

Having no history of any other diseases or syndromes.Having no history of any type of surgery.Having no malnutrition or obesity.Having no history of vitamin D supplementation.

In the control group (Group 2), we recruited healthy children from the local community aged 2 to 14 years. We obtained the serum vitamin D levels of children who met the control inclusion criteria. Group 2 included 275 children, but 13 blood samples were inadequate for vitamin D testing, so they were excluded, and only 262 children were included in the analysis.

#### 2.2.2. Inclusion Criteria for Group 2

5.Having no history of recurrent tonsillitis or any other diseases or syndromes.6.Having no history of any type of surgery.7.Having no malnutrition or obesity.8.Having no history of vitamin D supplementation.

### 2.3. Measurement of Serum Levels of Vitamin D 

Serum 25(OH)D levels were tested using a 5P02 ARCHITECT 25(OH)D Reagent Kit. It is a quantitative delayed one-step competitive immunoassay.

### 2.4. Interpretation of Serum Levels of Vitamin D 

Vitamin D deficiency occurs when levels fall below 20 ng/mL; insufficiency is between 20 and 29 ng/mL, and sufficiency is 30 ng/mL and above [9,10,11]. The concentration required to achieve an appropriate immunological effect is 20 to 50 ng/mL [23].

### 2.5. Factors Affecting Serum Vitamin D Levels

The associations between vitamin D level and each of the following factors were studied: age, sex, breastfeeding, formula-based or mixed feeding in infancy, mode of delivery, approximate daily milk intake, rate of tonsillitis, the presence of middle ear effusion, and snoring/mouth breathing, history of pathological neonatal jaundice, and sun exposure. All these factors were included in the analysis, except estimated daily milk intake and sun exposure because they were subject to significant estimation and recall bias.

### 2.6. Power Analysis

The power analysis with G*Power (version 3.1.9.3 for Mac. Erdfelder, Faul, & Buchner, Bonn, Kiel, Trier, Germany) using *t*-tests for independent groups showed a minimum sample of 470 participants (235 for each group) to achieve a power of 0.90, representing the chance of finding an association between study variables, and an effect size of 0.5 (medium to large effect), representing the likelihood of finding an association between the two independent samples with an alpha level of 0.05 (2-tailed), in order to reject the null hypothesis. A study sample of 504 participants were recruited, including 242 in the recurrent tonsillitis group (Group 1) and 262 in the control group (Group 2).

### 2.7. Statistical Analysis

Data were analyzed with the Statistical Package for Social Sciences (IBM SPSS statistics for Mac, version 25, Chicago, IL, USA). All data were assessed initially using a Kolmogorov–Smirnov test, histograms, and Q-Q plots to test for normality, followed by Levene’s test for equality of variances. Continuous variables were expressed as mean ± standard deviation and categorical data as frequencies and percentages. Correlations between continuous variables were analyzed using Spearman’s rank correlation test. Categorical variables were analyzed using an *X^2^*-test and continuous variables between groups with a *t*-test and one-way ANOVA. For unequal variances, non-parametric tests were used (Mann–Whitney U and Kruskal–Wallis tests). The results were regarded as statistically significant when the *p*-value was <0.05. After assumptions were met, a multivariable-adjusted Poisson regression was run to predict the number of tonsillitis episodes per year (count variable) based on vitamin D level (continuous variable).

## 3. Results

Group 1 consisted of 242 patients aged 2–14 years, with a mean age of 5.5 ± 2.2 years. Meanwhile, group 2 consisted of 262 participants in the age range 2–14 years with a mean age of 5.8 ± 2.3 years. No difference was found between the age means (*p* = 0.096). One-hundred-thirty (53.7%) and 138 (52.7%) participants were male in groups 1 and 2, respectively. No difference in sex distribution was observed between the groups (*p* = 0.814). No difference was found between body mass index (BMI) means for group 1 (16. 46 ± 1.1) and group 2 (16.63 ± 1.1). In group 1, the number of recurrent tonsillitis episodes per year ranged 3–12 (median = 6, IQR = 4). Ninety-six participants (39.7%) from group 1 had seven or more episodes of tonsillitis per year. The mean level of 25(OH)D in group 1 was 16.6 ± 7.3 ng/mL. One hundred seventy-five participants (72.3%) fell within the deficiency, 49 (20.3%) within the insufficiency, and 18 (7.4%) within the sufficiency range. The mean 25(OH)D level for group 2 was 26.1 ± 16.4 ng/mL. Ninety-one (34.7%) participants fell within the deficiency, 109 (41.6%) within the insufficiency, and 62 (23.7%) within the sufficiency range. The mean 25(OH)D level was significantly higher in group 2 than in group 1 (*p* < 0.0001) (Table 1).

The distribution of vitamin D levels of the two groups was represented using a Mann–Whitney U test. This showed that the median of Group 2 (22.8 ng/mL, IQR = 12.2) was significantly higher than the median of Group 1 (16 ng/mL, IQR = 9) *(p* < 0.001) (Figure 1).

Group 1 also demonstrated a significantly higher prevalence of Vitamin D deficiency and lower prevalence of vitamin D insufficiency and sufficiency compared to group 2 (all *p* values < 0.0001) (Figure 2).

Spearman’s rank correlation test showed no association between age and vitamin D level or rate of tonsillitis per year (*p* = 0.165 and 0.545, respectively). Age was also not correlated with vitamin D level in group 2 (*p* = 0.900). On the other hand, in both groups 1 and 2, BMI was correlated with vitamin D level (*p* < 0.0001, rho = −0.42 and *p* < 0.0001, rho = −0.3, respectively) and the rate of tonsillitis per year in group 1 (*p* < 0.001, rho = 0.24).

The prevalence of vitamin D level categories in group 1 across various demographic and clinical variables is represented in Table 2. The correlation between 25(OH)D deficiency and the following variables was studied: age groups (<7 and ≥7 years); sex; rate of tonsillitis (<7 and ≥7 episodes per year); mode of delivery; breast-, formula-, or mixed-feeding in infancy; presence of pathological neonatal jaundice; snoring and mouth breathing; and middle ear effusion (Table 2).

Spearman’s rank correlation test showed no association between age and vitamin D level or rate of tonsillitis per year (*p* = 0.165 and 0.545, respectively). Age was also not correlated with vitamin D level in group 2 (*p* = 0.900). On the other hand, in both groups 1 and 2, BMI was correlated with vitamin D level (*p* < 0.0001, rho = −0.42 and *p* < 0.0001, rho = −0.3, respectively), and the rate of tonsillitis per year in group 1 (*p* < 0.001, rho = 0.24).

The prevalence of vitamin D level categories in group 1 across various demographic and clinical variables is represented in Table 2. The correlation between 25(OH)D deficiency and the following variables was studied: age groups (<7 and ≥7 years); sex; rate of tonsillitis (<7 and ≥7 episodes per year); mode of delivery; breast-, formula-, or mixed-feeding in infancy; presence of pathological neonatal jaundice; snoring and mouth breathing; and middle ear effusion (Table 2).

In groups 1 and 2, a one-way ANOVA test showed no difference in vitamin D levels between age groups 0–5, 6–10, and 11–14 years (*p* = 0.867 and *p* = 0.471, respectively). In addition, an *X*^2^-test showed no significant difference in vitamin D deficiency between age groups <7 and ≥7 years (*p* = 0.532 and *p* = 0.803, respectively). Moreover, the independent samples *t*-test showed that the means of age were not statistically different in patients with or without vitamin D deficiency (*p* = 0.738 and *p* = 0.811, respectively). Furthermore, there was no difference in vitamin D deficiency between sexes (*p* = 0.149 and *p* = 0.425, respectively) (Table 2).

As also observed in Table 2, an *X*^2^-test showed that in group 1, vitamin D deficiency was significantly different between participants with ≥7 episodes of tonsillitis per year and those with <7 episodes (*p* < 0.0001). No difference in vitamin D level was found between patients born via C-section or normal vaginal delivery (*p* = 0.073).

Regarding the type of feeding during infancy in group 1, no difference in vitamin D deficiency was observed between those with breastfeeding or without (*p* = 0.958) or with a breastfeeding duration of <1 year and those breastfed for >1 year (*p* = 0.927). Furthermore, no difference in vitamin D deficiency was detected between those with formula feeding or without and with mixed feeding or without (*p* = 0.638 and 0.643, respectively). In addition, in group 1, no difference was observed in vitamin D deficiency between those with pathological neonatal jaundice, snoring and mouth breathing, and middle ear effusion or without (*p* = 0.578, 0.762, and 0.643, respectively) (Table 2).

In group 1, one-way ANOVA was utilized to compare rates of tonsillitis per year across vitamin D level categories and revealed a statistically significant difference (*p* < 0.0001). Multiple post hoc comparisons with Tamhane’s procedure showed that the mean in the deficiency category (7.7) was higher than in the insufficiency category (5.4) (*p* < 0.0001), and was also higher than in the sufficiency category (4.3) (*p* = 0.008) (Figure 3).

In group 1, a multivariable-adjusted Poisson regression of the rate of recurrent tonsillitis and vitamin D level (OR = 0.969 (95% CI, 0.962 to 0.975)) showed that for every single unit increase in vitamin D level, there was a 3.1% decrease in the number of tonsillitis episodes per year (*p* < 0.0001) (Table 3).

Finally, the correlation coefficient squared (R^2^ = 0.337) indicated that almost 34% of the variation in the rates of tonsillitis per year was due to the variation in vitamin D levels. The slope of the linear regression line clearly showed that the rate of tonsillitis decreased with increasing vitamin D levels (Figure 4).

## 4. Discussion

Tonsillitis is one of the main causes of hospital visits, and recurrent tonsillitis is the most common indication of tonsillectomy among children [22,24]. The exact etiology and pathogenesis of recurrent acute tonsillitis are still not fully understood [25]. Antibiotics often have low efficacy against recurrent tonsillitis, probably due to the formation of biofilms [6,26]. Vitamin D has been proposed to inhibit the biofilm formation associated with recurrent bacterial tonsillitis [6]. Such an association suggests that treating vitamin D deficiency may help lower rates of tonsillitis, tonsillectomies, and the expected postoperative complications [22].

In this study, our aim was to evaluate the association between vitamin D levels and the rate of recurrent tonsillitis in children. We compared vitamin D levels in a group of children with recurrent tonsillitis (Group 1) to a control group (Group 2). We also evaluated factors that might be associated with vitamin D level abnormalities such as age, sex, mode of delivery, feeding in infancy, snoring and mouth breathing, and middle ear effusion.

In both study groups, we observed that vitamin D status was mostly deficient or insufficient. Vitamin D level depends on the relative contribution of a wide range of environmental, constitutional, and genetic factors. Although our region is known for its generous months of sunshine, almost all year round, vitamin D deficiency or insufficiency is reported to be prevalent in young healthy Jordanian women [27]. Our findings were similar in one third of the study children of both groups. The reasons behind this include the conservative attire (owing to traditional values in Jordan), inadequate sun exposure behavior (hours and timing of exposure), and last, but not least, dietary intake [28]. Food rich in vitamin D includes oily fish, egg yolk, cod liver oil, and fortified vitamin D food, such as milk and cereal [27]. Unfortunately, most eating practices and habits in Jordan are heavily based on unbalanced meals, and fortified vitamin D products are not a primary concern. Other factors, including adiposity, age, sex, and various disease processes, can influence vitamin D levels [27].

BMI was negatively associated with vitamin D levels in both tonsillitis and control groups in our study. This finding is in line with previous studies [29] and could be explained by the fact that Vitamin D is fat-soluble and can be diffused into adipose tissue, causing serum levels to decrease [27].

Vitamin D levels in patients having recurrent tonsillitis were significantly lower than those of the control group. The prevalence of vitamin D deficiency was much higher in the tonsillitis group. Furthermore, a bigger number of patients showed vitamin D deficiency in the group with over seven episodes of tonsillitis per year compared to those with less than seven episodes per year. Our data demonstrate that the rate of tonsillitis per year increased with the decrease in vitamin D levels and vice versa. In our study, the findings were similar to the results of Yildiz et al. [4], where the average serum vitamin D levels were significantly lower in a group of children having recurrent tonsillitis compared to the control group.

Reid et al. [25] also suggested that 78% of children who underwent tonsillectomy had a 25(OH)D level <75 nmol/L. On the contrary, in a study conducted by Aydin et al. [27], no difference was found in mean vitamin D levels between children with recurrent tonsillitis and controls. However, the prevalence of vitamin D insufficiency was found to be higher in their tonsillitis group. Such findings were also observed in adult patients with recurrent streptococcal tonsillitis [30].

No difference was found in vitamin D between children of C-section or normal vaginal delivery. While our results contradict some reports in infants [31,32], they agree with the South Korean data association between vitamin d deficiency and primary cesarean section [33]. Since our study sample included the ages 2–14 years, the effect of mode of delivery is most likely masked, leaving other influential factors to predominate.

Feeding during infancy plays a vital role in determining vitamin D levels in childhood. No difference in vitamin D deficiency was found between children exclusively breastfed as infants and others. This can be explained by the vitamin D supplements given to most newborns up until the age of 6 months in Jordan. Additionally, no difference was found between formula or mixed feeding. This strengthens our hypothesis that other factors are perhaps of more value after the period of infancy. The findings of our study indicate that vitamin D deficiency could be regarded as a risk factor for recurrent tonsillitis, because of the resultant disruption of the role of vitamin D in modulating innate and adaptive immunity.

A main study limitation to note is that only one blood sample per subject was tested. It is, however, notable that Vitamin D is not a routine test, it being difficult to take multiple samples during the recruitment period of the study in a pediatric population. Furthermore, the retrospective nature of the study and the age distribution of the studied patients, could also be claimed as limitations to be taken into account.

## 5. Conclusions

In conclusion, our study is an important addition to what was previously studied regarding the effect of vitamin D on recurrent tonsillitis. Vitamin D deficiency is, in fact, significantly associated with the rate of recurrent tonsillitis. Future studies should concentrate on testing the role of Vitamin D supplements in protecting from recurrent tonsillitis and tonsillectomy.

## Figures and Tables

**Figure 1 ijerph-19-08744-f001:**
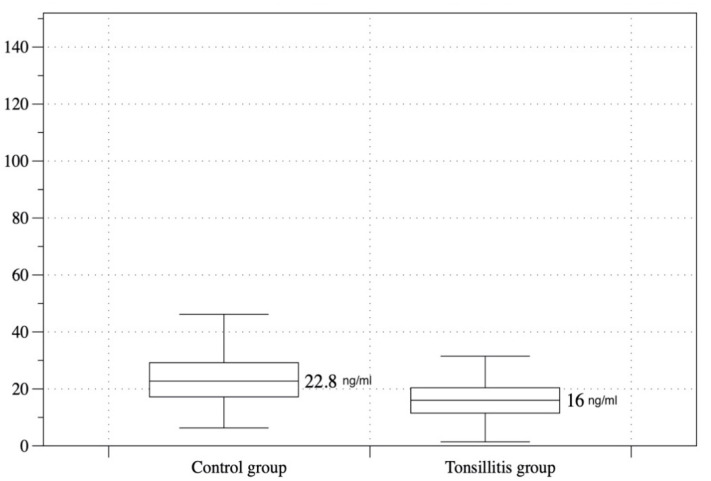
Distribution of serum 25-hydroxyvitamin D level in groups (box-and-whisker plot graph).

**Figure 2 ijerph-19-08744-f002:**
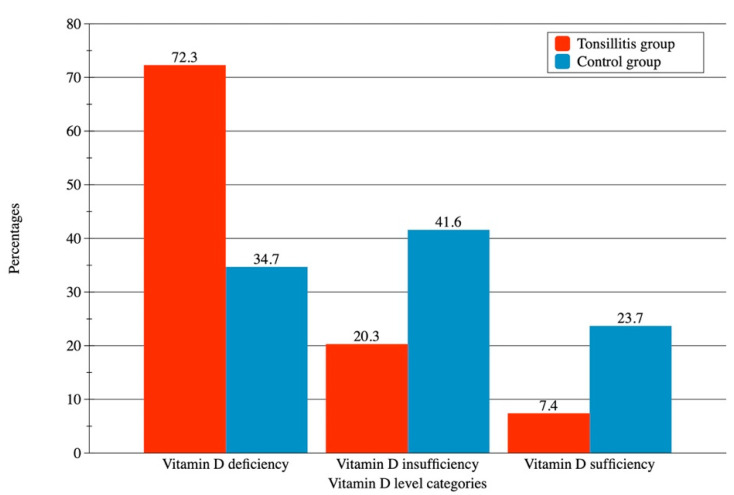
Percentages of participants in various vitamin D level categories in tonsillitis and control groups.

**Figure 3 ijerph-19-08744-f003:**
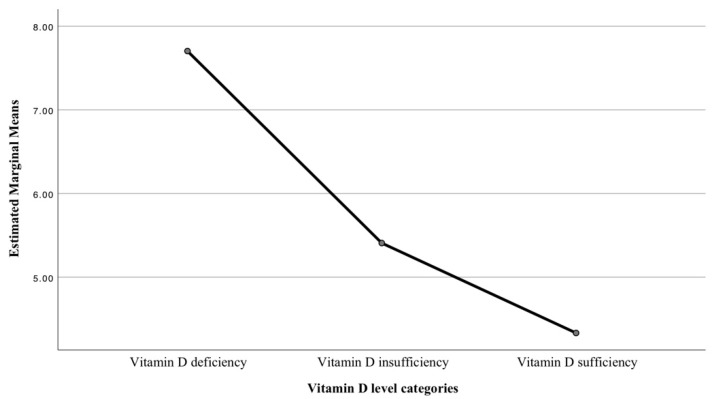
Estimated marginal means of the rate of tonsillitis per year.

**Figure 4 ijerph-19-08744-f004:**
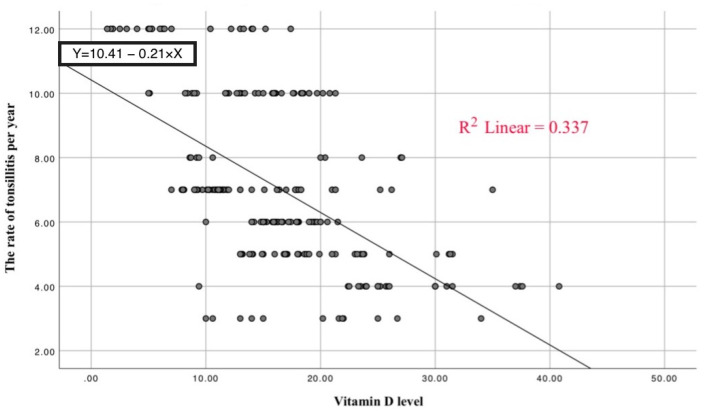
Scatter plot of vitamin D levels and the rate of tonsillitis per year.

**Table 1 ijerph-19-08744-t001:** Demographic and clinical characteristics of the study groups.

Characteristic	Cases (*n* = 242)	Controls (*n* = 262)	*p*-Value	
			X^2^-Test	*t*-Test
**Age**	5.5 ± 2.2	5.8 ± 2.3		0.096
**Male sex**	130 (53.7%)	138 (52.7%)	0.814	
**BMI**	16.46 ± 1.1	16.63 ± 1.1		0.074
**Serum 25-hydroxy Vitamin D level (ng/mL) mean ± SD**	16.6 ± 7.3	26.1 ± 16.4		<0.0001
**Deficiency (level < 20 ng/mL)**	175 (72.3%)	91 (34.7%)	<0.0001	
**Insufficiency (level 20–29 ng/mL)**	49 (20.3%)	109 (41.6%)	<0.0001	
**Sufficiency (level ≥ 30 ng/mL)**	18 (7%)	62 (23.7%)	<0.0001	

**Table 2 ijerph-19-08744-t002:** Prevalence of vitamin D level in patients with recurrent tonsillitis (group 1) correlated with demographic and clinical variables.

Characteristics							X^2^-Test	
							*p*-Value	φ
**Age**	˂7 years 181 (74.8%)			≥7 years 61 (25.2%)			0.532	
	Deficient	Insufficient	Sufficient	Deficient	Insufficient	Sufficient		
	129 (71.3%)	39 (21.6%)	13 (7.1%)	46 (75.4%)	10 (16.4%)	5 (8.2%)		
**Sex**	Male 130 (53.7%)			Female 112 (46.3%)			0.149	
	Deficient	Insufficient	Sufficient	Deficient	Insufficient	Sufficient		
	89 (68.5%)	31 (23.8%)	10 (7.7%)	86 (76.8%)	18 (16.1%)	8 (7.1%)		
**Rate of tonsillitis**	˂7 117 (48.3%)			≥7 125 (51.7%)			<0.0001	0.362
	Deficient	Insufficient	Sufficient	Deficient	Insufficient	Sufficient		
	65 (55.6%)	35 (29.9%)	17 (14.5%)	110 (88%)	14 (11.2%)	1 (0.8%)		
**Mode of delivery**	Normal vaginal 170 (70.2%)			Cesarean section 72 (29.8%)			0.073	
	Deficient	Insufficient	Sufficient	Deficient	Insufficient	Sufficient		
	115 (67.7%)	43 (25.3%)	12 (7%)	60 (83.4%)	6 (8.3%)	6 (8.3%)		
**Breast feeding in infancy**	No 7 (2.9%)			Yes 235 (97.1%)			0.958	
	Deficient	Insufficient	Sufficient	Deficient	Insufficient	Sufficient		
	3 (42.8%)	2 (28.6%)	2 (28.6%)	172 (73.2%)	47 (20%)	16 (6.8%)		
	˂1 year 142 (58.7%)			≥1 year 100 (41.3%)			0.927	
	Deficient	Insufficient	Sufficient	Deficient	Insufficient	Sufficient		
	104 (72.7%)	31 (21.7%)	8 (5.6%)	71 (71.7%)	18 (18.2%)	10 (10.1%)		
**Formula feeding in infancy**	No 39 (16.1%)			Yes 203 (83.9%)			0.638	
	Deficient	Insufficient	Sufficient	Deficient	Insufficient	Sufficient		
	29 (74.4%)	3 (7.7%)	7 (17.9%)	146 (71.9%)	46 (22.7%)	11 (5.4%)		
**Mixed feeding in infancy**	No 46 (19%)			Yes 196 (81%)			0.643	
	Deficient	Insufficient	Sufficient	Deficient	Insufficient	Sufficient		
	32 (69.6%)	5 (10.9%)	9 (19.6%)	143 (73%)	44 (22.4%)	9 (4.6%)		
**Pathologic neonatal jaundice**	No 216 (89.3%)			Yes 26 (10.7%)			0.578	
	Deficient	Insufficient	Sufficient	Deficient	Insufficient	Sufficient		
	155 (71.8%)	48 (22.2%)	13 (6%)	20 (76.9%)	1 (3.9%)	5 (19.2%)		
**Snoring and mouth breathing**	No 30 (12.4%)			Yes 212 (87.6%)			0.762	
	Deficient	Insufficient	Sufficient	Deficient	Insufficient	Sufficient		
	21 (70%)	9 (30%)	0 (0%)	154 (72.6%)	40 (18.9%)	18 (8.5%)		
**Middle ear effusion**	No 168 (69.4%)			Yes 74 (30.6%)			0.643	
	Deficient	Insufficient	Sufficient	Deficient	Insufficient	Sufficient		
	120 (71.4%)	35 (20.8%)	13 (7.8%)	55 (74.3%)	14 (18.9%)	5 (6.8%)		

Results are expressed as *n* (%); NS: not significant; φ: strength and direction of association of *X*^2^-test.

**Table 3 ijerph-19-08744-t003:** Poisson regression of vitamin D level with potential confounding factors and the rate of tonsillitis per year.

Parameter	B	Std. Error	95% Wald Confidence Interval		Hypothesis Test			OR	95% Wald Confidence Interval for OR	
			Lower	Upper	Wald Chi-Square	df	*p*-Value		Lower	Upper
**(Intercept)**	2.798	0.4574	1.901	3.694	37.407	1	<0.0001	16.406	6.694	40.214
**Age**	0.002	0.0153	−0.028	0.032	0.025	1	0.873	1.002	0.973	1.033
**Sex**	0.047	0.049	−0.049	0.143	0.906	1	0.341	1.048	0.952	1.153
**BMI**	−0.024	0.0307	−0.084	0.036	0.597	1	0.440	0.977	0.920	1.037
**Vitamin D level**	−0.032	0.0036	−0.039	−0.025	78.105	1	<0.0001	0.969	0.962	0.975

Dependent Variable: Rate of Tonsillitis per year, Model: (Intercept), Age, Sex, BMI, Vitamin D Level.

## Data Availability

Not applicable.

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
