# Peer review of "Vitamin D Levels in Children with Recurrent Acute Tonsillitis in Jordan: A Case-Control Study"

_ijerph, 2022, doi:10.3390/ijerph19148744_

Round 1
Reviewer 1 Report
I do not know whether the references are set according to the journal's instructions. They seem mixed up. They are not numbered either according to the order referred in the text nor according to name order
They are also mixed up in the discussion as pointed below.
lines 235- 237: The whole sentence should be rephrased and cut into two. Like : Although our region is known for its generous months of sunshine almost all year round, vitamin D deficiency or insufficiency is reported to be prevalent in young healthy Jordanian women [37]. Similar were our findings in one third of the study children of both groups.
Line238:......, inadequate sun exposure...The word inadequate could be added
line 241: the two references 41& 42 are irrelevant in that phrase context, since the papers do not refer to anything about food vitamin D content.
Lines 246 : I think the reference 46 is irrelevant. BMI connection with vitamin D is not mentioned in that paper. Anyhow it is referred to infants and low D levels. Or up to the reference 46 the sentence should be rephrased. The reference 44 I think is relevant to BMI and D
line 255. Yildiz et al is the 41 reference and not 43. The reference 42 is also relevant here although in adults where that should be pointed out.
Line 258: Reid et al is ref 43 and not 45
line 259-261: Reference 28 Aydin et al...... the context of the sentence is not reported exactly. They say that mean D levels did not differ significantly between controls and the recurrent tonsillitis group. The word mean I think should be added.
Line 263: I think the correct ref is 42 and not 44
Where are the references 48 & 49?
The ref 47 refers to connection of women's cesarean section and vitamin D levels
Where are the Korean data you refer to?
The ref 45 concerns infants in Korea aged 1-6 months with prevalence in D deficiency
Anyhow all the references need reordering properly to be able to make sense.
Author Response
lines 235- 237: The whole sentence should be rephrased and cut into two. Like : Although our region is known for its generous months of sunshine almost all year round, vitamin D deficiency or insufficiency is reported to be prevalent in young healthy Jordanian women [37]. Similar were our findings in one third of the study children of both groups.
Response: We appreciated the reviewer's comment. This was addressed and changes were implemented in the text a long with next suggestion.
Line238:......, inadequate sun exposure...The word inadequate could be added
Response: Addressed. Please check the attached manuscript.
line 241: the two references 41& 42 are irrelevant in that phrase context, since the papers do not refer to anything about food vitamin D content.
Response: These two references were deleted.
Lines 246 : I think the reference 46 is irrelevant. BMI connection with vitamin D is not mentioned in that paper. Anyhow it is referred to infants and low D levels. Or up to the reference 46 the sentence should be rephrased. The reference 44 I think is relevant to BMI and D
Response: Addressed. Please check the attached manuscript.
line 255. Yildiz et al is the 41 reference and not 43. The reference 42 is also relevant here although in adults where that should be pointed out.
Response: Addressed. Please check the attached manuscript.
Line 258: Reid et al is ref 43 and not 45
Response: Addressed. Please check the attached manuscript.
line 259-261: Reference 28 Aydin et al...... the context of the sentence is not reported exactly. They say that mean D levels did not differ significantly between controls and the recurrent tonsillitis group. The word mean I think should be added.
Response: We thank the reviewer for her/his comment. The word "mean" was added.
Line 263: I think the correct ref is 42 and not 44
Response: Addressed. Please check the attached manuscript.
Where are the references 48 & 49?
Response: Added, 32 & 33 respectively.
The ref 47 refers to connection of women's cesarean section and vitamin D levels
Response: The reference has been changed accordingly
Where are the Korean data you refer to?
The ref 45 concerns infants in Korea aged 1-6 months with prevalence in D deficiency
Response: The reference has been changed accordingly
Anyhow all the references need reordering properly to be able to make sense.
Response: References were all reordered using Mendeley.

Reviewer 2 Report
- The IRB number should be added in line 74.
- The effect size seems too small. Can you explain it?
- Please unify the writing of p value format.
- The title of the result is missing.
- Table 1,2,3 format needs to be confirmed.
- Table1 needs to add the comparative difference of the two groups of covariates.
- The study flow chart provides a better understanding of the study design.
- Confirmation of the unit of the Figure ordinate
- What are the strengths of this study?
- Are there any limitations to this study?
Author Response
Comments and Suggestions for Authors
- The IRB number should be added in line 74.
Response: Addressed. Please check the attached manuscript.
- The effect size seems too small. Can you explain it?
Response: Apologize, 0.5 medium to large effect size was used but 0.3 written by mistake.
- Please unify the writing of p-value format.
Response: Addressed. Please check the attached manuscript.
- The title of the result is missing.
Response: Please note the title was added.
- Table 1,2,3 format needs to be confirmed.
Response: Tables were modified based on the journal requirement.
- Table1 needs to add the comparative difference of the two groups of covariates.
Response: In table 1 the comparative difference is provided for covariates along with used statistical test and P values
- The study flow chart provides a better understanding of the study design.
Response: Thank you for suggesting the flow chart. However, we think that the study design is clearly described in the methodology part. We will be happy to do further clarification for any point if needed.
- Confirmation of the unit of the Figure ordinate.
Response: Addressed. Please check the figure in the attached manuscript.
- What are the strengths of this study?
Response: We believe that the strength of this study is being the first study on Jordanian children to study the relation between Vitamin D and recurrent tonsillitis.
- Are there any limitations to this study?
Response: One main limitation is that only one blood sample was tested per subject. Vit D is not a routine test and it was difficult to take multiple samples during the recruitment period of the study. The other important point is the retrospective nature of the study and the high age range of the recruited patients. Although we believe that our sample size was representative, larger sample size can be advantageous. However, due to the pandemic, it was not easy to recruit more patients for this study.

Reviewer 3 Report
General comments:
This is an interesting manuscript and overall well written with minimal concerns regarding flow or grammar.
It sounds like recurrent tonsillitis was diagnosed based on recall. Is it possible to clarify whether episodes of tonsillitis were physician-diagnosed or reported by families without confirmation?
In Figure 1, although there was a significant difference reported from the statistical analyses the box-and-whisker plot also shows overlap between the two groups.
It appears that only one blood sample for vitamin D was taken for each participant. The outcome was the number of episodes of tonsillitis per year, but it was not clear for each participant how many years were included, or whether this was a lifetime report where the number of years would vary by child age. Modalities of feeding, such as breastfeeding, were included, but because age varied from 2-14 years with a mean age of 5 years, it is not clear whether this would have had a current impact on vitamin D levels in older children. It will be helpful if the authors can comment on the temporality of testing vs episodes of tonsillitis and clarify how the number of episodes per year were examined in this cohort (e.g. how many years were considered, etc). While the authors do address potential reasons for low vitamin D levels detected in their study population, a comment on the likelihood that these single samples are representative for the years of the study will be helpful.
Specific comments:
Line 74: IRB number is not provided
Line 115: Do the authors refer to the biologic sex of the child or the child's preferred gender? For a study like this, biologic sex would likely be of more scientific utility than gender. Lines 174, 184, 230 also refer to gender.
Line 214: due to or positively associated with? Bradford-Hill criteria for attribution do not appear to be clearly met.
Author Response
This is an interesting manuscript and overall well written with minimal concerns regarding flow or grammar.
It sounds like recurrent tonsillitis was diagnosed based on recall. Is it possible to clarify whether episodes of tonsillitis were physician-diagnosed or reported by families without confirmation?
Response: We thank the reviewer for her/his raising this point and yes. Tonsillitis was the clinical diagnosis that was documented in the patients' medical record.
In Figure 1, although there was a significant difference reported from the statistical analyses the box-and-whisker plot also shows the overlap between the two groups.
Response: Yes, we observed the overlap in the box-and-whisker plot but since the test showed statistically significant differences, we reported it as such.
It appears that only one blood sample for vitamin D was taken for each participant. The outcome was the number of episodes of tonsillitis per year, but it was not clear for each participant how many years were included, or whether this was a lifetime report where the number of years would vary by child age. Modalities of feeding, such as breastfeeding, were included, but because age varied from 2-14 years with a mean age of 5 years, it is not clear whether this would have had a current impact on vitamin D levels in older children. It will be helpful if the authors can comment on the temporality of testing vs episodes of tonsillitis and clarify how the number of episodes per year was examined in this cohort (e.g. how many years were considered, etc). While the authors do address potential reasons for low vitamin D levels detected in their study population, a comment on the likelihood that these single samples are representative of the years of the study will be helpful.
Response: We agree with the reviewer on this comment. Yes, only one blood sample was taken for each participant because there were no readings for vit D levels in patient medical records. Vit D is not a routine test for such patients and it was difficult to take multiple samples during the recruitment period of the study. This is a limitation of this study and this is expected with such a study design. We acknowledged this within the limitations of this study.
Regarding the reporting of tonsillitis, yes, this was a lifetime report where the number of years varied by child age, however, the mean age was 5 years.
Regarding breastfeeding and its impact, perhaps the remote history might be the reason for not finding a statistically significant association.
Specific comments:
Line 74: IRB number is not provided:
Response: Please note the IRB number has been added
Line 115: Do the authors refer to the biologic sex of the child or the child's preferred gender? For a study like this, biologic sex would likely be of more scientific utility than gender. Lines 174, 184, 230 also refer to gender.
Response: Addressed. We meant by gender the biologic sex, therefore the word gender was replaced with the word sex.
Line 214: due to or positively associated with? Bradford-Hill criteria for attribution do not appear to be clearly met.
Response: We thank the reviewer for her/his excellent remark. Maybe the Bradford Hill criteria were not clearly met but we think that not all of these criteria are applicable to this study due to its retrospective nature. That is why we prefer to refrain from using them in the manuscript in order not to confuse the readership of our study. We hope that the reviewer considers that discussing each criterion separately and whether it is applicable or not might result in a long discussion and increase the word count limit. We believe that we reported the results of our analyses clearly and we hope that it is clear for our readers.

Round 2
Reviewer 2 Report
Thanks for the reply, but I don't seem to see any supplements for limitations in the revised article.
Author Response
Based on your suggestion, we added the following paragraph at the end of the discussion.
One main limitation is that only one blood sample was tested per subject. Vitamin D is not a routine test and it was difficult to take multiple samples during the recruitment period of the study. The other important point is the retrospective nature of the study and the high age range of the recruited patients.